

# Last Glacial Maximum and Deglacial Abyssal Seawater Oxygen Isotopic Ratios

Carl Wunsch[1]

[1,*]Department of Earth and Planetary Sciences, Harvard University
[*]Also at Department of Earth, Atmospheric and Planetary Sciences, MIT

*Correspondence to:* Carl Wunsch (cwunsch@fas.harvard.edu)

**Abstract.** An earlier analysis of pore water salinity/chlorinity in two deep-sea cores, using terminal constraint methods of control theory, concluded that although a salinity amplification in the abyss was possible during the LGM, it was not required by the data. Here the same methodology is applied to $\delta^{18}O_w$ in the upper 100m of four deep-sea cores. An ice volume amplification to the isotopic ratio is, again, consistent with the data but not required by it. In particular, results are very sensitive, with conventional diffusion values, to the assumed initial conditions at -100 ky. If the calcite values of $\delta^{18}O$ are fully reliable, then inferred enriched values of the ratio in sea water are necessary to preclude sub-freezing temperatures, but the sea water $\delta^{18}O$ in pore waters does not independently support the conclusion.

## 1  Introduction

Based upon the work of McDuff (1985), Schrag and de Paolo (1993), Schrag et al. (2002), Adkins and Schrag (2001, 2003), Insua et al. (2014), and several others, on the properties of pore-waters in abyssal cores, inferences have been made about the salinity and temperature of the regional and global abyssal oceans during the last glacial maximum (LGM). A summary of the central conclusion (e.g. Adkins et al., 2002) would be that the ocean was almost everywhere near freezing temperatures, and with an abyssal salinity much above the global volume average, particularly in the Southern Ocean.

Those inferences have become a part of the ongoing discussion of climate physics, including the whereabouts of global carbon during the LGM, and are a standard against which models are being tested: e.g., Otto-Bliesner et al. (2006), IPCC AR5 (2013), Kobayashi et al. (2015). Recently Miller (2014), Miller et al. (2015) have challenged this interpretation showing, using a Monte Carlo method, that the uncertainties of the inferences were too great to assert that the LGM abyssal stratification could be determined with useful accuracy.

Their conclusion was tested by Wunsch (2015; hereafter W15) using salinity (chlorinity) data obtained from the pore-waters of two of the cores used by Adkins and Schrag (2003) and Miller (2014), Miller et al. (2015; hereafter M15). In contrast to the latter authors, the analysis was carried out in the physically more direct context of standard control theory: pore fluid data were treated as a "terminal constraint" on the time-evolving pore-water properties.[1] Using highly optimistic assumptions (a

---

[1]Miller et al. (2015) used a Markov Chain-Monte Carlo (MCMC) approach. Whether this stochastic method is intuitively more accessible than the one used here is a matter of taste.





known one-dimensional advection-diffusion model with perfect parameters, known initial conditions, etc.) the uncertainties in the estimated abyssal salinity through time supported the Miller et al. (2015) inference. In general, the very high local values of abyssal salinity, $S$, much above the LGM volumetric mean, were possible within the uncertainties of the chlorinity/salinity data and its model, but were not required by the data and model.

The purpose of this present paper is to extend the W15 salinity analysis to the pore-water measurements of the oxygen isotope ratio, $\delta^{18}O_w$; see Schrag and dePaolo (1993), Adkins and Schrag (2001, 2003), Schrag et al. (2002), M15. The oxygen isotope tracer is of particular importance for the interpretation of the calcite ratio of $\delta^{18}O_c$ in foraminifera, to separate the ice volume effect (controlling $\delta^{18}O_w$) from the temperature signature in $\delta^{18}O_c$ during the last glacial maximum. As discussed by these previous authors, both colder in situ water temperatures, and increases in global ice volume lead to an increase in

$\delta^{18}O_c$ (cf. Bradley, 1999, p. 199+). Unless the ice volume contribution in $\delta^{18}O_w$ is sufficiently large, some $\delta^{18}O_c$ values imply sub-freezing deep ocean temperatures. Schrag et al. (2002) discuss the limits on the required global volumetric mean increase in $\delta^{18}O_w$, with a lower bound (Duplessy, 1978; Bradley, 1999) of 1.1 o/oo to prevent below-freezing temperatures as inferred from $\delta^{18}O_c$. A maximum change of 1.3 o/oo is found if the entirety is attributed to a sea level–drop of about 120m at the LGM (Fairbanks, 1989) with no associated temperature change. But as Schrag et al. (2002) and others emphasized, no reason

exists to believe that any kind of spatially uniform changes occurred during the glacial and deglacial intervals. The temperature change estimate leans on finding the small difference between two noisy numbers and involves the accuracy of the average global ice sheet $\delta^{18}O$ during the LGM and non-negligible salinity effects in $\delta^{18}O_c$ among other problems.

As in W15, the focus here is on the upper 100m of the cores, where the observed $\delta^{18}O_w$ has its maximum, instead of aiming for an overall analysis from the full pore-water data depth range. Again the emphasis is on understanding the extent to which

the $\delta^{18}O_w$, by themselves, imply large ambient values in the abyssal waters.

The general procedure here for both salinity and $\delta^{18}O_w$ is identical to that in W15 which has a broader discussion. That is, any tracer $c(z,t)$ in the pore-waters is supposed to satisfy a one-dimensional advection-diffusion equation,

$$\frac{\partial c}{\partial t} + w(z,t)\frac{\partial c}{\partial z} - \frac{\partial}{\partial z}\left(k(z,t)\frac{\partial c}{\partial z}\right) = \tag{1a}$$

$$\frac{\partial c}{\partial t} + \left(w(z,t) - \frac{\partial k(z,t)}{\partial z}\right)\frac{\partial c}{\partial z} + k(z,t)\frac{\partial^2 c}{\partial z^2} = 0. \tag{1b}$$

Here $z$ is positive upwards from the base of the core data, time runs forward, $w$ is an advective flow relative to the surrounding solid, but porous, medium and $k$ is a vertical diffusion coefficient. No diagenetic reactive processes are included (see e.g., Berner, 1980; Schrag and de Paolo, 1993). $\partial k/\partial z$, if non-zero, can be thought of as an effective velocity, $w^*$, but it competes with the effects of variable $k$ in the last term of Eq. (1b). Derivation of this equation is not straightforward, and involves numerous assumptions discussed by Berner (1980), Boudreau (1997), Huettel and Webster (2001), Bruna and Chapman (2015),

Voemans et al. (2016) and others. Because this equation has been the model used by previous workers, it is simply adopted here as a black-box framework for discussion of the resulting uncertainties in the pore-water inferences with little discussion of its



probable violations. $\delta^{18}O_w$ is a *ratio* of concentrations, but the denominator (the concentration $\left[^{16}O\right]$) is treated as invariant so that a standard advection-diffusion equation is still appropriate.[2]

If $k, w$ are known, Eq. (1) is a conventional parabolic partial differential equation in $z, t$ whose textbook solution involves specifying: (a) the initial conditions, (b) the boundary condition at $z = h$ (the fluid-solid interface) and (c) the boundary condition at $z = 0$, the base of the pore-water data. As discussed by Wunsch (2006), problems involving observations almost never coincide with the well-posed situations described in most differential equations textbooks, and one must specifically ask "what is known, and what has to be determined?" In the present situation with real core data, apart from the model, information is available only at the time when the core was obtained, $t = t_f$, and includes $c(z, t_f) = C(z, t_f) \pm \Delta C(z, t_f)$ where $C(z, t_f)$ are the "true" terminal values, and $\Delta C(z, t_f)$ is an estimate of their accuracy. In many cores, the porosity and tortuosity of the solid phase are also measured at time $t_f$ and are used to infer $k(z, t_f)$, also with some (not always stated), uncertainty. The measured values, $C_{term} = c(z, t_f)$, are the "terminal constraint" on the tracer, and represent the values that any estimate of the core values through time, $c(z, t)$, $0 \leq t < t_f$ to which the concentration must converge within error bars (here also labelled "uncertainty").

Because the core properties are measured only at time $t_f$, the initial conditions, $C_0(z)$, are unknown and a plausible guess is required. Two candidates suggest themselves: (1) Relying on the rough quasi-periodicity of Pleistocene ice ages (see e.g., M15), one can set $C_0(z) = c(z, t_f) \pm \Delta C(z, t = 0)$, but where $\Delta C(z, t = 0) >> \Delta C(z, t = t_f)$ owing to the significant uncertainty. (2) Take the completely agnostic value $C_0(z) = 0 \pm \Delta C(z, t = 0)$, and attempt to determine the initial conditions from the data. (See Table 1 for the notation.)

What of the boundary condition $C_h(t) = C(z = h, t)$? Determining $C_h(t)$ has been the focus of the existing literature, with inferred values being the estimates of LGM and deglacial abyssal salinities and isotopic ratios and is what led to the general inferences described above. As with all estimation problems, it helps greatly to have a good (accurate) a priori estimate of the true value. As outlined by McDuff (1985) and Adkins and Schrag (2001), plausibly both oceanic salinity and $\delta^{18}O_w$ would be controlled largely by ice volume changes, and they and subsequent investigators have usually taken a scaled version of the estimated sea level curve as a sensible starting place. Because sea level curves are sometimes based upon $\delta^{18}O$ values in corals and cores and which are presumably correlated with $\delta^{18}O_w$, we follow M15, in using their estimate—one that avoided such data and as shown in Fig. 1, scaled for $\delta^{18}O_w$. An enrichment at the estimated global ice maximum about -20 ky BP occurs. The curve, which will be used as an a priori estimate, was allowed to range from -0.19 to 1.16 o/oo with a mean value of 0.43 o/oo.

Let the guessed prior be written as $\tilde{C}_h(t) = C_h(t) \pm \Delta C_h(t)$, where $\Delta C_h(t)$ is another uncertainty estimate. The bottom boundary condition is problematic. With finite diffusion and/or an upward directed $w$, structures below the measured pore fluid depth can propagate into the measured domain. A separate or combined calculation would be required for determining those unknown structures, and we again follow previous investigators in using a zero-flux boundary condition at $z = 0$. The problem can easily be reformulated to determine that boundary condition instead of, or in addition to, $C_h(t)$, and noting that in general, $\partial c(0)/\partial z \neq 0$ in the data.

---

[2]This assumption would be a poor one for tracers, such as isotopic ratios of $Nd$, where the denominator can vary significantly over the ocean volume.



## 2    Estimation Structure

Following W15, the problem is written in discrete numerical form using a Dufort-Frankel method (Roache, 1976) with the exception that now $k$ will be treated explicitly as a linear function of $z$ with a factor of two variation, but it makes little quantitative difference to the results; see the Appendix). W15 described three, equivalent, stable methods for solving the resulting

terminal constraint estimation problem.[3] Again, with the goal of finding the most optimistic estimates of uncertainty, $w, k$ are treated here as perfectly known, all data being used to compute the control variable, which is the correction to the guessed a priori $C_h(t)$ and its uncertainty. Here only the so-called RTS smoother algorithm is used, as it produces rigorous uncertainty estimates (rigorous up to the model choice, including the prior uncertainties). With $k, w$ known and time-independent, the problem is a linear one, with the model written in "state-space" form as,

$$\mathbf{x}(t+\Delta t) = \mathbf{A}(t)\mathbf{x}(t) + \mathbf{B}(t)\mathbf{q}(t) + \mathbf{\Gamma}(t)\mathbf{u}(t), \ t = 0, \Delta t, 2\Delta t, ..., (N-1)\Delta t, \tag{2}$$

$$t_f = (N-1)\Delta t$$

$\mathbf{A}$ is the $2M \times 2M$ "state transition" matrix (a function of $w, k$), $\mathbf{q}(t)$ represents the prescribed boundary conditions and $\mathbf{B} = \mathbf{\Gamma}$ distributes the boundary condition over the requisite grid points.[4] Here $\mathbf{q} = q(t)$ is a scalar, and $\mathbf{B}$ is a vector of all zeros except with unity at the top boundary point–the core-ocean interface or,

$$\mathbf{B} = \mathbf{\Gamma} = [0, 0, ...0, 0, 1]^T, \tag{3}$$

that is, zero vectors except for the last point. The state vector, and hence the terminal data has dimension $2M$, where $M$ is the number of vertical gridpoints in $z$. $\mathbf{u}(t)$ is the "control" and is the adjustment that will be made to $\mathbf{q}(t)$ to render the state as consistent as possible with the terminal data conditions. The state vector is $\mathbf{x}(t) = [c(\mathbf{z}, t - \Delta t), c(\mathbf{z}, t)]$ with discretized vector $0 \le z \le L$, in $\Delta z$. The numerical scheme requires two time levels in $\mathbf{x}(t)$. In the present special case, matrices, $\mathbf{A}, \mathbf{B}, \mathbf{\Gamma}$ are

here all taken to be time-independent and perfectly known, and $\mathbf{q}(t) = q(t)$, $\mathbf{u}(t) = u(t)$ are scalars representing the abyssal water $\delta^{18}O_w$ prior, and adjustment respectively. (Many extensions of this formalism exist, including non-linear systems.)

An equation governing the observations, $\mathbf{y}(t)$, is written,

$$\mathbf{E}(t)\mathbf{x}(t) + \mathbf{n}(t) = \mathbf{y}(t), \tag{4}$$

where $\mathbf{n}(t)$ is the noise in the observations. In the present special case, $\mathbf{E}(t) = 0$, $t < t_f$, $\mathbf{E}(t_f) = \mathbf{I}_{2M}$ where $\mathbf{I}$ is the identity

matrix (that is, observations exist only at the terminal time).

The estimation equations are described more fully in W15. Full specification of the system includes these equations plus all of the a priori estimates of uncertainty in parameters, initial conditions, measurement noise, Etc.

---

[3]Anyone interested in the rigorous mathematics of such problems in continuous space and time is urged to consult Lions (1971).

[4]The notation is that bold lower-case letters indicate column vectors; bold upper-case letters (Latin or Greek) are matrices, and the superscript $T$ means the transpose. Vectors and matrices sometimes reduce to scalars.





## 2.1 Identification

Standard control theory and statespace methods (e.g., Goodwin and Sin, 1984; Franklin and Powell, 1998; Wunsch, 2006) commonly distinguish between two problems associated with Eqs. (2,4), that of "identification" and "statespace and control estimation." The identification problem in this case reduces to answering the question: "What values of the parameters $w, k$

are the best ones to use in modelling the data?" The formalism following from Eq. (2) is sufficiently general to include representations of the space-time structure of $k(z, t_f), w(z, t_f)$—if equations governing their time-space evolution were available. Absent such information, the simplest, but arbitrary and optimistic, assumptions range from assuming constant values in one or both of space and time to, at the opposite extreme, assuming a white-noise structure in both space and time ending with the structures observed at the terminal time in the core. The former assumptions under-parameterize the true variability, and

the latter introduce an enormous number of further unknown parameters relative to the available data. Various intermediate assumptions can be made.

## 2.2 State Estimation and Control

If the identification problem has been solved, producing a useful model (or "plant" in the engineering control literature), available data can be used instead to determine the adjusted boundary condition $\tilde{C}_h(t) = q(t) + \tilde{u}(t)$, and the best estimate of the

full state variable $\tilde{\mathbf{x}}(t) \to [c(\mathbf{z}, t)]$. It was this second problem addressed in W15—in which guesses were made of the most appropriate model and the terminal constraint problem then solved by standard sequential methods (Lagrange multipliers/adjoint, and the Rauch-Tung-Striebel (RTS) smoother). It was argued then, that the uncertainty of the resulting $\tilde{C}_h(t)$ was so great, despite using all of the terminal data to determine it, that little could be said about the abyssal water salinity change during the LGM and the subsequent deglaciation. Using some or all of the same data to *also* determine $k(z, t), w(t)$ could only further

*increase* the uncertainty of the estimate $\tilde{u}(t)$. This result was consistent with M15.

## 2.3 Observability and Controllability

Control methods introduce the concepts of observability and controllability (Wunsch, 2006; Marchal, 2014) as well as a series of related ideas such as "reachability" (see Goodwin and Sin, 1984). Here, "observability" means that the observations are adequate to perfectly reconstruct the initial conditions. "Controllability" implies that the system can be driven from any initial

condition to an arbitrary terminal value.

The extent to which the terminal data are determined by the initial conditions is an important issue here. Thus (e.g., Wunsch, 2006, p. 233) with a single observation at the end time, and in the absence of any external disturbance, the observability matrix is

$$\mathbf{O} = \mathbf{I}_{2N} \mathbf{A}^{t_f} = \mathbf{A}^{t_f},$$

and with $C_h(t) = 0$ would, if $\mathbf{A}^{t_f}$ is of full rank, permit exact solution of

$$\mathbf{x}(t_f) = \mathbf{O}\mathbf{x}(0),$$





for $\mathbf{x}(0)$. Loss of information about the initial conditions will arise directly from the dissipative nature of diffusion or, if there is a finite $w$, from the sweeping out of information by advection from the region of observation. Using $k = 10^{-10}$ m$^2$/s, $w = 0$, $\Delta t = 127.3$ y, and $t_f = 786\Delta t$ y, $L = 100$ km and 101 grid points in $\mathbf{z}$, the rank of $\mathbf{O}$ is 24 ($\mathbf{A}$ has rank 100, the number of non-surface-boundary grid points in the vertical). Thus a "range" of 24 structures in the initial conditions can be

inferred from the terminal data, and 76 will lie in its null space. With these parameters, the system is not fully observable and the question is whether the null space is of serious concern or not. (Structures in the terminal state null space of $\mathbf{O}$ are not determined by the initial conditions, and might be provided by the control instead. Structures in its range *can* be provided by the initial conditions, but can also be provided by the control.) Loss of information between the starting and ending times is intuitively sensible: small vertical-scale structures in $\mathbf{x}(0)$ do not survive measurably over 100,000 ky in a diffusive system.

Large vertical-scale structures can and do survive; see the analytical solutions in W15 and the cases analyzed below.

Suppose that the initial condition were zero. Then "controllability" would answer the question of whether any choice of control in $C_h(t) = q(t) + u(t)$ would carry the system to the terminal data $C_{term}(z)$? Then the controllability matrix $\mathbf{\Theta}$ (e.g., Wunsch, 2006, p. 232) is

$$\mathbf{\Theta} = \left\{\mathbf{I}_{2N}, \mathbf{A}, \mathbf{A}^2, ..., \mathbf{A}^{t_f - \Delta t}\right\}\mathbf{\Gamma} \qquad (5)$$

The system is controllable only to the extent that $\mathbf{\Theta}$ is full rank, $M$, for $t_f - \Delta t = (N-2)\Delta t$. In the present case, from the definition of $\mathbf{\Gamma}$ (Eq. 3), the rank is estimated as about 33.

Neither of these concepts depends on the actual data. The formalisms can be used to find explicit descriptions of the terminal data structures determinable from the initial conditions and controls. Here we proceed instead by direct construction of the solutions, having inferred that there will be a strong dependence on *both* initial conditions and controls, with some inevitable

residuals to be regarded as "noise." A fuller discussion of controllability and observability depends upon understanding whether the smaller, but non-zero, eigenvalues of $\mathbf{A}$ and its powers are sufficiently large compared to the unclear noise level.

## 3 The Data

Fig. 2 displays the positions of the five cores for which $\delta^{18}O_w$ data were available (courtesy of M. Miller, personal communication, 2015, and see Table 1) superimposed upon the *modern* $\delta^{18}O_w$, distribution at 3500m from the GISS website; see

LeGrande and Schmidt (2006). The modern range at this depth is roughly from -0.3 to 0.3 o/oo, plus outliers. An artificial boundary for the Antarctic-origin bottom waters, owing to a lack of data, is visible (see LeGrande and Schmidt, 2006), as is the relatively strong gradient in the Atlantic Ocean. Any calculated global spatial average from four locations for this or any other depth would have a large uncertainty. For reference purposes, a straight area-weighted average of the gridded values in Fig. 3 is -0.013 o/oo.

Measured terminal porosity in each of the cores is displayed in W15. Fig. 3 shows the $\delta^{18}O_w$ measurements with depth, with the exception of core 1239, which is shown in Fig. 4.

The visible fluctuations in all cores exceed the estimated analytical accuracy of 0.03 o/oo (Adkins and Schrag, 2001), but the extent to which they represent real changes in boundary conditions through time, their initial conditions and fluxes from below





| Core No. | Reference | Location | Water Depth (m) |
|---|---|---|---|
| ODP981 | Jansen et al. (1996) | NE Atlantic, Feni Drift/Rockall | 2200 |
| ODP1063 | Keigwin et al. (1998) | Bermuda Rise | 4600 |
| ODP1093 | Gersonde et al. (1999) | Southern Ocean, SW Indian Ridge | 3600 |
| ODP1123 | Carter et al. (1999) | E. of New Zealand, Chatham Rise | 3300 |
| ODP1239 | Mix et al. (2002) | E. Tropical Pacific, Carnegie Ridge/Panama Basin | 1400 |

**Table 1.** Cores from which chlorinity/salinity data were used, along with a reference to their initial description in the Ocean Drilling Program (ODP) and with a geographical label. A nominal water depth of the core-top is also listed.

the measured core depth, as opposed to a variety of noise processes in the formation of a core undergoing active sedimentation, remains obscure. One of the major issues is whether structures other than the visible overall maximum, presumably at the LGM, are signals to be understood, or mere noise, to be suppressed.

Differences among the core $\delta^{18}O_w$ do not easily support an hypothesis of any kind of globally uniform variation in the bottom water concentration, $C_h(t) = q(t) + u(t)$ anywhere below about 100m depth. Visually, cores 1093 and 1239 are qualitatively different from the other three, with 1239 showing very large excursions near-surface, and 1093 having roughly constant values with depth, but with superimposed structures. Under-sampled core 1239 has extreme values represented primarily by single point excursions, likely connected to the extreme volatility of dynamical properties in the equatorial Pacific Ocean and is not further discussed here. The remaining three cores all have a visual maximum of greater or lesser definition at some tens of meters depth. Whether other features are noise or signal is an imponderable. Differences in core water depths must always be borne in mind as well.

Differences among the cores imply that there need not be any overall, that is global, control on their time histories. (See the cautionary statements in Schrag et al., 2002). Dynamics and modern oceanographic structures (as in Fig. 2) instead support the accepted inference of different time histories of the values of $\delta^{18}O_w$ in the bottom waters, consistent with the different core profiles.

Fig. 5 shows normalized versions of the oxygen isotope and salinity data in the cores. If these two properties satisfy the same advection-diffusion equation (1), they must have different temporally varying boundary and/or initial conditions, or be subject to possible biogeochemical interactions not treated here.

Begin as in W15, in which the observed core provides the terminal constraint, and the initial condition is assumed to be the same as the terminal one, but with a larger error estimate. Absent any more compelling possibility, the same sealevel curve, is used, but scaled to lie between -0.2 o/oo and 1.15 o/oo (Fig. 1).

## 3.1 The Average $\delta^{18}O_w$ Core

Knowledge of oceanic dynamics and the modern distribution, as well as the core $\delta^{18}O_w$ data in Fig. 3 make it very unlikely that a globally uniform shift in the oxygen isotope ratio ever occurred. Injection of ice-melt, precipitation, and evaporation necessary





to remove and create continental ice sheets controlling the porewater $\delta^{18}O_w$ involve primarily oceanic surface properties, and the time scale to reach any kind of dynamic and kinematic equilibrium over the entire ocean volume requires thousands of years (e.g., Wunsch and Heimbach, 2008; Siberlin and Wunsch, 2010; Gebbie, 2012).

In the absence of regionally coherent core porewater data, a major problem is determining the extent to which structures in the $\delta^{18}O_w$ data represent purely local "noise", or regionally important climate signatures that must be understood. As a simplified context for later discussion of the individual cores, a start is made by averaging the four cores displayed in Fig. 3, with the result also shown there, and extending to the depth of the shallowest record (138m). An average core does not exist in nature, but provides a generic data set to discuss the methodology and results. In any core, one can guess at the structures to be treated as a noise process rather than as signal. Averaging is a primarily data-based noise reduction process, in which incoherent small-vertical scale features will tend to be suppressed. With only four examples, the standard error of the result, shown in Fig. 6, is very large, having only three degrees of freedom. Nonetheless, we proceed. No attention has been paid to differences in sedimentation rate, or other depth controlling processes. Results will be used as a framework for later discussion of the individual cores. Because of the linearity of the problem, the final estimation uncertainties do not depend upon the data themselves. In addition, the control solution for the average core will be the same as averaging the controls of the individual cores—if the same statistics are used for them.

The analysis follows much of the earlier literature in setting $w = 0$. Results from assuming a purely diffusive response, "near-periodic" initial conditions (initial conditions set to the terminal data), and a constant $C_h(t) = 0$ are shown in Fig. 7. $\mathbf{P}_0 = (2\text{o/oo})^2\,\mathbf{I}$, $Q = (2\text{ o/oo})^2$. A very large uncertainty, $\pm 2$ o/oo is assigned to both initial conditions and the control, respectively. The terminal data uncertainty is the calculated standard deviation of the four cores. Unless specifically stated otherwise, $k$ is linear over the top 100m core depth with values $5 \times 10^{-11} - 10^{-10}\text{m}^2/\text{s}$ in all cases.

The fit to the terminal state is statistically acceptable, with an isotopic maximum at 60-70 m. On the other hand, no significant LGM maximum appears in the control—instead, the smoother places most of the structure into the initial conditions—which, consistent with the observability discussion, persists as a local maximum through the 100 ky time interval. This result emphasizes the ambiguity of initial conditions and control with pure diffusion at the assumed rate over both 100m in the vertical and 100 ky over the time duration.

Examples such as this one render concrete a number of interlocking elements of the problem. (1) Noise or uncertainty covariances for the initial conditions, the terminal data, and the prior $C_h(t)$ are as much a part of the model as is $k$, and the underlying partial differential equation, or the data themselves. Their choices determine what is regarded as signal and what is noise. (2) Consider an extreme case. By setting the initial conditions equal to the terminal data within some uncertainty, and letting $k \to 0$, a completely acceptable solution would be found by fixing the control as the constant $C_h(t) = C_{term}(z = 0)$. In the limit, all of the initial condition structure is maintained through to $t_f$. The only reason to preclude such a solution is the requirement that $k > 0$. (3) In producing a local maximum at the depth inferred to be the properties of the LGM, the governing equation produces a tight tradeoff between a large value of $k$, permitting adequate penetration to the observed depth, versus its strong tendency to diminish the amplitude of the resulting maximum (cf. Adkins and Schrag, 2003). Whether both amplitude





and depth can be simultaneously reproduced, with a simple rule for $k$, has to be determined in each case, and a judgment may have to be made as to which, if either feature, is the more robust element in the data?

In contrast to the quasi-periodic initial and final conditions, (Fig. 8) shows the result when the initial condition was taken to be zero: the system responds by reconstructing the near-periodic initial condition. Again the residual is acceptable, and the

control hardly differs. Initial conditions are very important with this diffusivity values and time interval.

Now consider what happens when the prior is taken to be the scaled sea level curve of Fig. 1, with zero initial conditions, and as shown in Fig. 9. The terminal fit is once again acceptable, and the control adjustments are very small. The standard inference of enhanced $\delta^{18}O_w$ at about -20 ky by order 1 o/o relative to today is also consistent with the model and the data. As in W15 for salinity, an LGM maximum is not *required* by them, but becomes an assumption to rationalize temperature data.

The strong dependence upon the initial condition is striking. *It can be suppressed* as shown in Fig. 10 where the initial condition was set to zero, with a minute uncertainty, and the terminal uncertainty was strongly downweighted in the vicinity of the depth of the local maximum. Then as shown in the figure, a solution reproducing the terminal maximum is found, with a time-varying control over almost the entire record. This solution is most like earlier published ones. On the other hand, the uncertainty in $\tilde{u}(t)$ still greatly exceeds any useful range.

When the sealevel prior is used in this situation (not shown), the final total control is visually very similar to that shown in Fig. 10. This result suggests that suppression of the initial conditions as unknowns brings the system closer to producing a unique control, but a zero initial state is not easily justified.

## 3.2 Core 1063

None of the five cores is obviously "typical", but core 1063 on the Bermuda Rise, a focus of the study of Adkins and Schrag

(2001), has the characteristic maximum at depth with a quasi-linear decrease with depth. Again only the upper 100m are considered.

The a priori terminal constraint variance is $\mathbf{R} = (0.3 \text{o/oo})^2 \mathbf{I}$, (about 10 times larger than the value in Adkins and Schrag, 2001) ($\mathbf{I}$ is the identity matrix and now meant to be approximately the analytical measurement error, $\mathbf{P}_0 = (0.17 \text{ o/oo})^2 \mathbf{I}$, $Q = (1.0 \text{ o/oo})^2$ somewhat more physically realistic than the very large, agnostic values used with the mean core. Only a

fraction of the total number of possible examples will be displayed.

Fig. 11 shows the purely diffusive solution with the sea level prior and the quasi-periodic initial condition. The region of influence of the core-top boundary condition is confined to about the top 50m, consistent with W15. Qualitative deviations occur only in the last 5000 years. This solution must be rejected as 90% of the terminal values lie outside the approximate 95% confidence (two-standard deviations) interval. Note again that none of the adjustments, $\tilde{u}(t)$, are statistically significant.

When $k$ was reduced by a factor of 100 below the nominal value used for the mean core, a better fit was found, but it still had to be rejected as the $\delta^{18}O_w$ maximum occurred about 10m above that observed in the core. It would appear that this core is not consistent with the prior within the stipulated error bars.





A possibility is that including a non-zero lower boundary condition, $c(z = 0, t)$, representing upward diffusion of signals from below would improve the residual—but no compelling reason exists for permitting that further increase in unknown degrees-of-freedom.

Fig.12 shows the Core 1063 solution with quasi-periodic boundary conditions when it is forced to the water column maximum by greatly reducing the estimated error in its vicinity, and with the sea level prior. The fit near the maximum is, as forced and expected, good, but the smaller scale structures are not reproduced. The same situation, but with the flat zero-prior is shown in Fig. 13. This solution is marginally better than for the sea level prior, but no LGM maximum appears in the control. In terms of the residuals, this solution effectively treats all structures in the top 100 m of the core, except for the maximum excursion, as a noise process. If that inference is accepted, then a posteriori, an estimate of the variance of the noise structure in the core has been made.

### 3.3 Core 1093

Core 1093, in the Southern Ocean on the Southwest Indian Ridge, has been the main basis of the inference of a strongly salinity stratified abyssal ocean during the LGM. For the top 100 m of $\delta^{18}O_w$, Fig. 14 shows that once again the quasi-periodic initial conditions with a flat prior can reproduce the terminal constraint but with no requirement of a maximum in $q(t) + u(t)$. The initial condition carries most of the structure.

### 3.4 Cores 981, 1123

Similar results emerge from the remaining two cores and so only representative solutions are shown in Figs. 15, 16, both for the case of quasi-periodic initial conditions, the flat prior and forcing to the $\delta^{18}O_w$ peak at depth. As in the other cores, the dependence on the initial conditions is clear. Both show a significant adjustment $\tilde{u}(t)$ near the terminal time, but neither requires an LGM peak, although the sea level prior is also acceptable (not shown).

## 4 Discussion

To a very great extent, the results of analyzing these cores depend very directly upon a long list of assumptions of which a rough summary if those made here would include:

(1) Physics/chemistry are one-dimensional

(2) Sedimentation rates are constant

(3) Rules for diffusivity/porosity/tortuosity are accurate

(4) Advection/diffusion without chemical reaction processes is adequate

(5) Initial conditions are similar to the terminal measurements, but with a larger uncertainty.

(6) $k(z, t), w(z, t)$ are time independent (and equal to the estimated terminal value).

(7) Structures in the terminal values are/are not signals (are/are not noise) and accuracy is dominated by analytical accuracy (or not)





(8) Lower boundary condition at $z = 0$ is one of no flux, with no upward diffusion or advection from below the data depth.

(9) The scaled sea level curve is a useful prior boundary condition estimate of order $\pm 1$ g/kg for salinity and +/- 1 o/oo for $\delta^{18}O_w$ (the latter is sometimes increased to $\pm 2$ o/oo to permit nearly arbitrary behavior).

(10) Variance estimates for the uncertainties in the terminal data and in the initial conditions are approximately correct.

One of the reasons this problem is so interesting is its intimate connection between the physical model (advection/diffusion), and the statistical model (the uncertainty estimates) and which govern the division between signal and noise in the data. For the range of $k$ used here, the competition between dominance by the initial conditions and the changes induced by the control can produce a realistic maximum at depth *only by interpreting everything except the gross shape as an unexplained noise structure*. This inference may well be correct, but is not proven. The great sensitivity to initial conditions can be reduced, as in W15, by

assuming a significant downward-directed advection velocity, $w < 0$. Support for such an hypothesis would have to come from a great deal more knowledge of the fluid-sediment dynamical model.

The overall inference here, consistent with both M15 and W15, is that the conventional picture of a very cold, highly saline abyssal ocean during the LGM remains possible, but is not a requirement of the existing data. If LGM $\delta^{18}O_w$ is insufficiently enhanced, then taken at face value, the $\delta^{18}O_c$ data would imply some oceanic temperatures below the freezing point. That issue

might be sufficient to be convincing evidence that high $\delta^{18}O_w$ *must* have occurred, but the dependence upon the reliability of the interpretation of the foraminifera data will be plain (e.g., Bradley, 1999). A growing literature (e.g., Marchal and Curry, 2008; Huybers and Wunsch, 2010; Burke et al., 2011; Gebbie, 2012; Amrhein et al., 2015; W15; M15) attempting to quantify inferred circulation differences between the LGM and the modern increasingly finds it difficult to distinguish any qualitative or even quantitative differences. Such findings do not disprove hypotheses of major change to water mass volumes, including

cold, strong abyssal salinities during the LGM, but only reinforce the need for far more data than are now available—if the hypotheses are to become factual.

## Appendix A:  Appendix: Numerical Tests

Consider a purely diffusive system, with $k = k_0 + k_1 z$, that is growing linearly from the bottom of the core. The governing equation is

$$(k_0 + k_1 z)\frac{\partial^2 c}{\partial z^2} + k_1 \frac{\partial c}{\partial z} - \frac{\partial c}{\partial t} = 0, \tag{A1}$$

subject to the periodic boundary condition, $C_h(t) = c(z = 0, t) = \cos(\sigma t)$, $\sigma = 2\pi/20000$y, with $t$ in years. Setting $c = \hat{c}(z)\exp(-i\sigma t)$, and making the substitution, $\zeta = k_0 + k_1 z$, a form of Bessel's equation is found (Olver, 2010)

$$\frac{d^2\hat{c}(z)}{dz^2} + \frac{1}{\zeta}\frac{d\hat{c}(z)}{dz} + \frac{i\sigma\hat{c}(z)}{k_1^2\zeta} = 0,$$

with solution

$$\hat{c}(\zeta) = aJ_0\left(\frac{2\sqrt{i\sigma}}{k_1}\zeta\right) + bY_0\left(\frac{2\sqrt{i\sigma}}{k_1}\zeta\right) \tag{A2}$$





where $J_0, Y_0$ are the Bessel functions, noting the singular behavior as $k_1 \to 0$ (Kelvin functions can also be used.) The upper and lower boundary conditions are

$$aJ_0\left(\frac{2\sqrt{i\sigma}}{k_1}\left(k_0 + k_1 L\right)\right) + Y_0\left(\frac{2\sqrt{i\sigma}}{k_1}\left(k_0 + k_1 L\right)\right) = 1, \; z = L$$

$$aJ_1\left(\frac{2\sqrt{i\sigma}}{k_1}k_0\right) + Y_1\left(\frac{2\sqrt{i\sigma}}{k_1}k_0\right) = 0, \; z = 0$$

5    and solved for $a, b$, where the identities $J_0' = -J_1$, $Y_0' = -Y_1$ were used. Fig. 17 shows the comparison between the numerical and analytical solutions, as well as the result with constant $k$ in place of the linear form used in this paper.

*Acknowledgements.* Supported in part by the National Science Foundation under Grant OCE096713 to MIT. I am again grateful to Dr. M. Miller for making the data available to me and for discussions of their use. D. Schrag was very helpful in skeptical discussions of the background, and of the details of this problem.



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



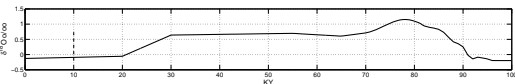

**Figure 1.** Prior estimate of the ocean-sediment interface boundary condition on $\delta^{18}O_w$, derived from the sea level curve of Miller et al. (2015). The maximum a priori value, for the LGM is 1.15±1o/oo and the minimum, calculated very approximately from the data in Fig. 2 is -0.2±1o/oo, consistent with previous such estimates in the references (e.g., Miller et al., 2015). Time zero is at -100 ky. Vertical dashed line indicates a range of 1 $o/oo$ between the nominal modern and the change to the LGM described by Schrag et al. (2002).

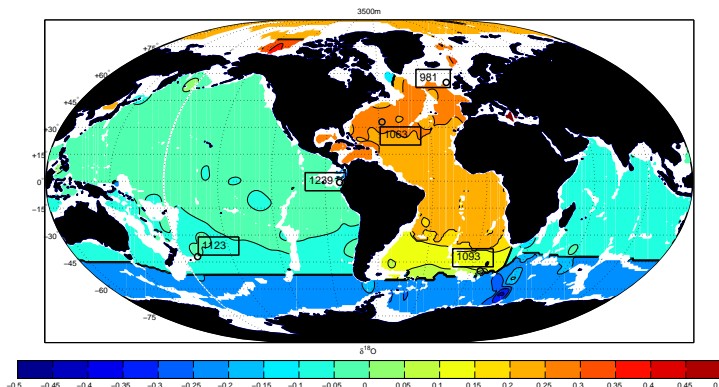

**Figure 2.** Modern $\delta^{18}O_w$ at 3500m (from LeGrande and Schmidt, 2006) with superimposed core positions (black circles). Note, however, that the core tops are not generally at this depth (Table 1). Note also (G. Gebbie, private communication, 2015) that the structures in this chart may be overly sensitive to the method used for gridding. Region of little or no data in the Southern Ocean is bounded by the thick black line.





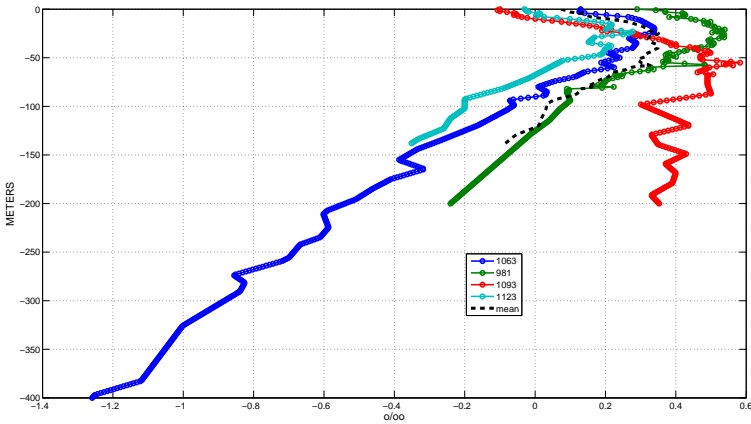

**Figure 3.** The four $\delta^{18}$O profiles used here along the mean value to the depth of the shallowest record (core 1123). Recall that the tops of the cores lie at different water depths.

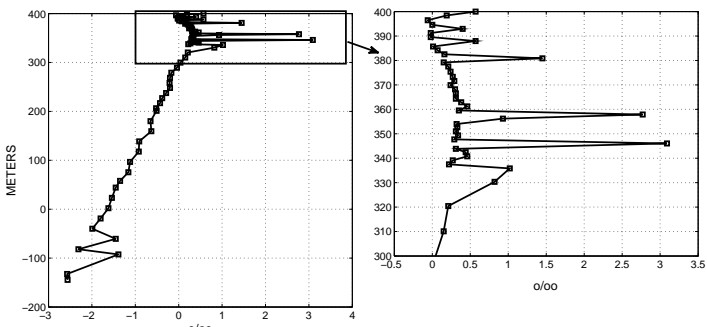

**Figure 4.** $\delta^{18}$O in core 1239, eastern equatorial Pacific, showing apparent undersampling near the core top and likely related to the intense dynamical variations expected on the equator and continental margins. Note the zero depth at 400m. These data were not further used here.

| Notation | Variable | Definition |
|---|---|---|
| Initial Condition | $C_0(z)$ | $c(t=0,z)$ |
| Boundary Condition | $C_h(t)$ | $c(t,z=0)$ |
| Terminal Condition | $C_{\text{term}}(z)$ | $c(t=t_f,z)$ |

**Table 2.** Notation used for initial, final and boundary conditions. In the discrete form, two time-steps of the concentration $c$ make up the state vector, boldmath$x(t)$, and corresponding imposed conditions, Tildes over variables denote estimates.




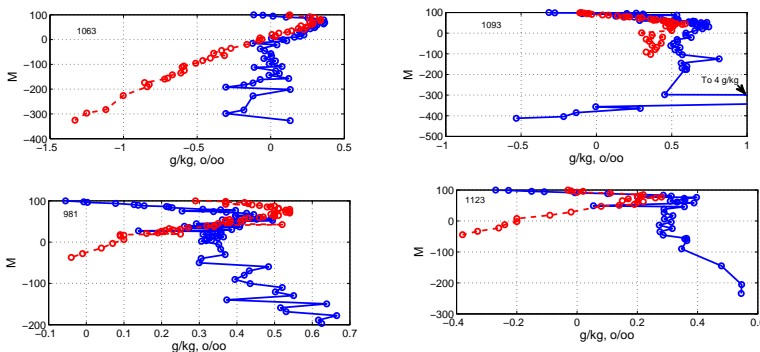

**Figure 5.** Measured salinity minus 35 g/kg (blue solid curves) and $\delta^{18}O_{sw}$ (red dashed) in four cores. With the possible exception of core 1093 in the upper 100m, the two data sets are quite distinct and hence inconsistent with a common advection-diffusion equation and boundary and initial conditions. The zero is set arbitrarily at 100m.

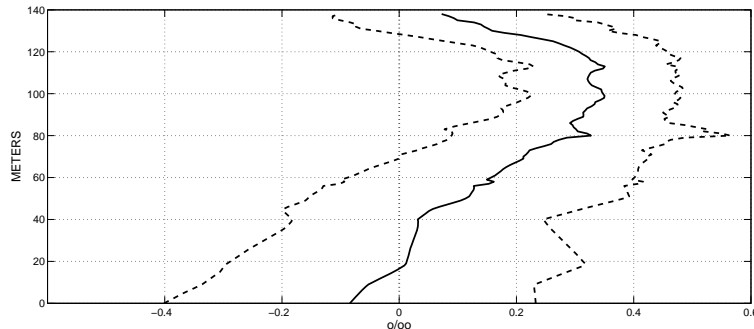

**Figure 6.** The mean of four core $\delta^{18}O$ values and their formal standard deviation with three degrees of freedom. Values differ from zero at one standard deviation only in the interval from about 10 to 70 m depth. Only the top 100 m of data are used in the analysis here.




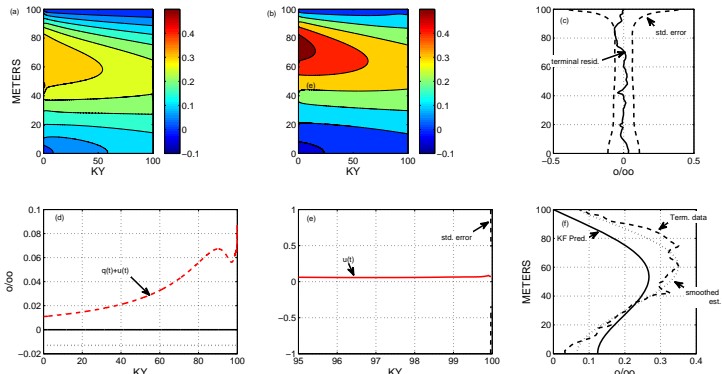

**Figure 7.** Purely diffusive solution for the mean core, to 100 m depth, with $k$ linearly increasing from $5 \times 10^{-11}$ m$^2$/s to $10^{-10}$ m$^2$/s, $w = 0$. The prior boundary condition is $C_h(t) = 0$ and the initial condition is the same as the terminal data, with larger uncertainties. Final values are all within one standard deviation of the estimated error bar. Apart from a small increase in $u$ (correction to $C_h(t)$ with time), the system reproduces the terminal constraint largely by adjusting the initial conditions. (a) Kalman filter (KF) solution from initial conditions and zero control adjustment. (b) Final smoothed estimate over the 100 ky. (c) Residual at the terminal time and one-standard deviation uncertainty limits. (d) The total control $q(t) + u(t)$ (dashed) and prior $q(t)$ (solid) as well as the estimated maximum and minimum of LGM $\delta^{18}O_{sw}$. (e) $u(t)$ except only the last 5 ky and showing the sharp drop in its uncertainty near the terminal time. Standard errors lie off-scale except at the very end. (f) Kalman filter (KF) prediction of the terminal data (same as a conventional forward calculation from the initial conditions and a priori $C_h(t)$, solid line), the terminal data (dash-dot), and the RTS fit to the data.

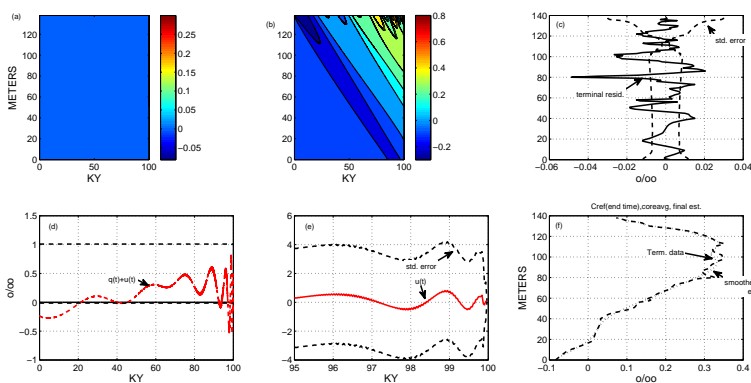

**Figure 8.** Same as Fig. 7 except that the initial condition was zero with a large uncertainty, rendering the Kalman filter solution zero until the very end. The smoothed solution is very similar to that with a near-periodic initial condition.




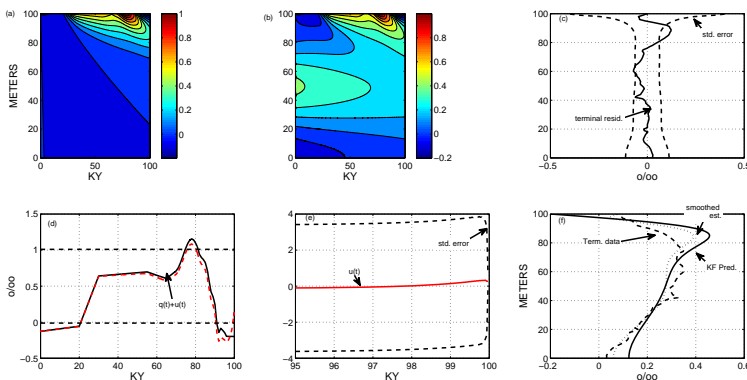

**Figure 9.** The mean core, upper 100m, zero initial conditions but with the sea level prior in Fig. 1. The small adjustment, $u(t)$, demonstrates that the terminal data are also consistent with the inference of a very high $\delta^{18}O_w$ at about -20ky with an increment exceeding 1 o/oo between the LGM and the present. Note, however, the very large uncertainties remaining in $\tilde{u}(t)$.

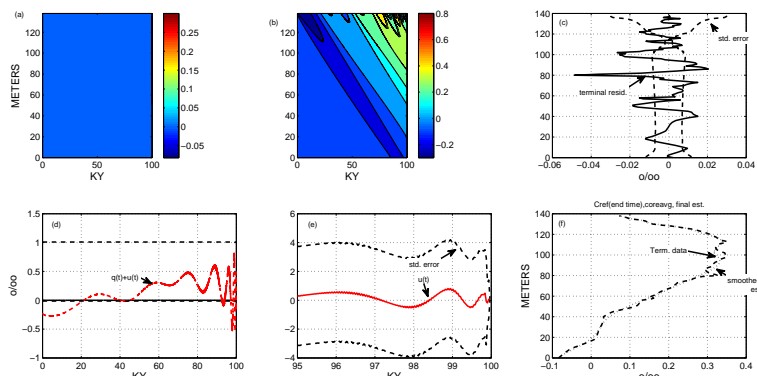

**Figure 10.** Using the full 140m of the average core, with the initial conditions prevented from changing significantly, and a flat prior, a solution reproducing the $\delta^{18}O_w$ maximum at depth through the reduced values of $\mathbf{R}$ there, and treating all other features as errors. $\tilde{u}(t)$ is now distributed over most of the 100 KY, but with no statistical significance anywhere. Strong reduction in the prior uncertainty of the terminal state is visible as the very narrow standard errors in the estimate visible in (c) and in the terminal reproduction of the gross maximum. Large number of outliers in $\tilde{c}(z, t_f)$ would imply an inconsistency between the solution and the prior error estimate.




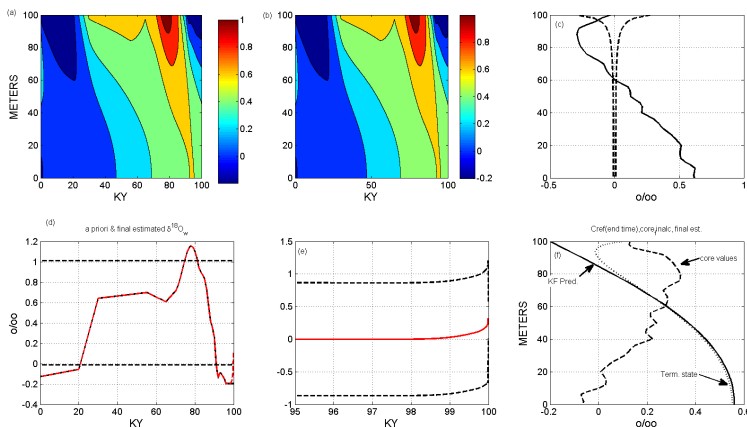

**Figure 11.** Results for core 1063 (Bermuda Rise) $\delta^{18}O_{sw}$, for a purely diffusive system, the sealevel prior, and quasi-periodic initial-conditions in a failed solution. 90% of the terminal misfit lies outside two standard deviations. The results of the Kalman filter are shown (a), and below about 50 meters in the core are strongly dependent upon the initial conditions which coincide with those of the core. Also shown is the misfit at the last time step, $t = t_f$ (model fit to the core) and the standard error (b). Adjustment to the a priori curve is very small except very close to the termination. Control $\tilde{u}(t)$ is shown (e) and which differs visibly from the prior only in the sharp upturn at the very end.

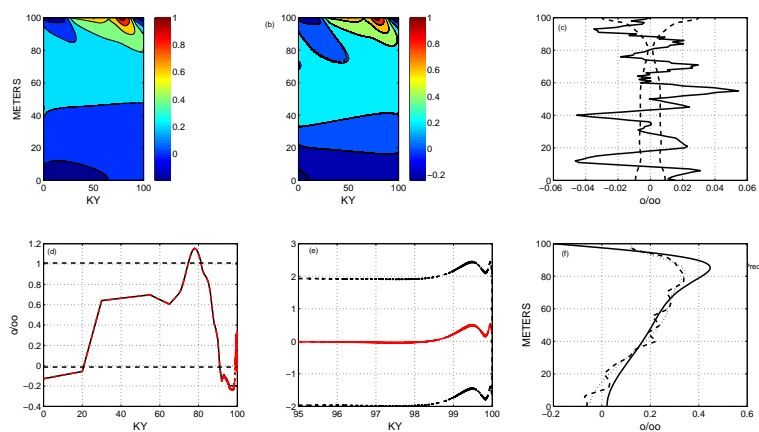

**Figure 12.** Same conditions as in Fig. 11 for core 1063 but with the solution forced to reproduce the local maximum through the terminal uncertainty estimate. All residual structures would be noise of unknown nature. The $C_h(t)$ maximum does exceed the estimated volumetric global mean, but the uncertainties remain close to $\pm 2$ o/oo. Again, pinched error bars in (c) show the forcing to the local maximum visible in (f). This solution is most like those discussed in the earlier literature, fitting only to the local maximum, and treating all other structures as noise.





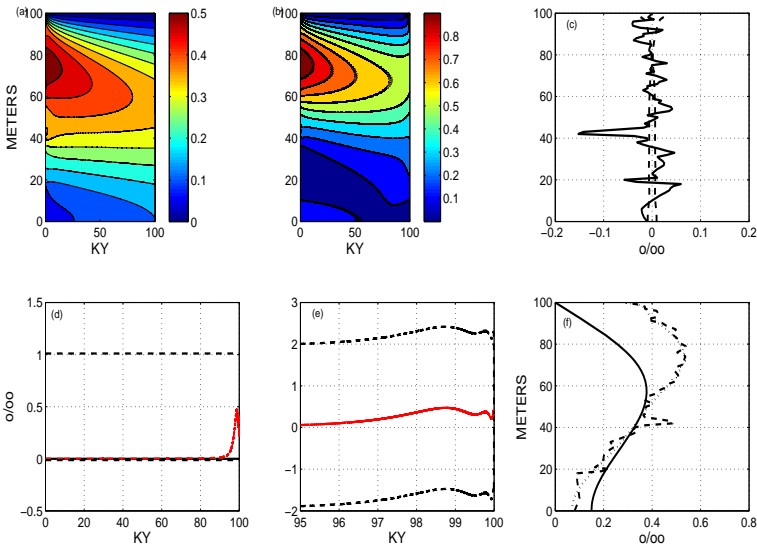

**Figure 13.** Core 1063 with a flat prior, quasi-periodic initial-final conditions, and with the terminal data uncertainty matrix **R** structured to emphasize the range of depths of the core maximum . Position and magnitude of the maximum are good. All other structures are then inferred to be a noise process. Compare Fig. 13

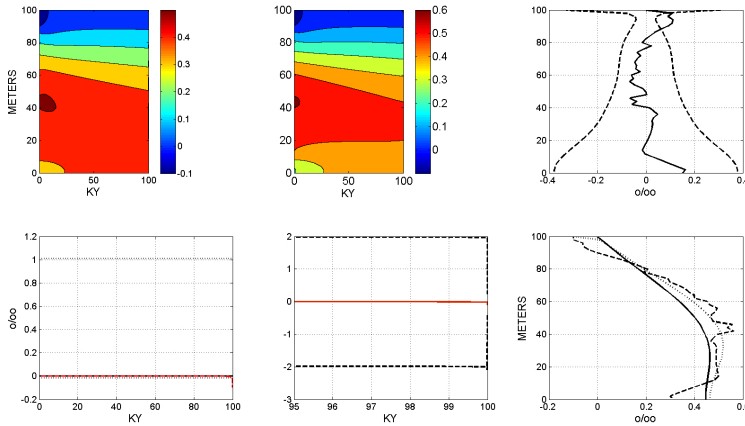

**Figure 14.** Core 1093 with a quasi-periodic initial condition, a flat prior, and terminal uncertainties forcing solution to the $\delta^{13}C_w$ at depth. Total control is flat, until the very end.





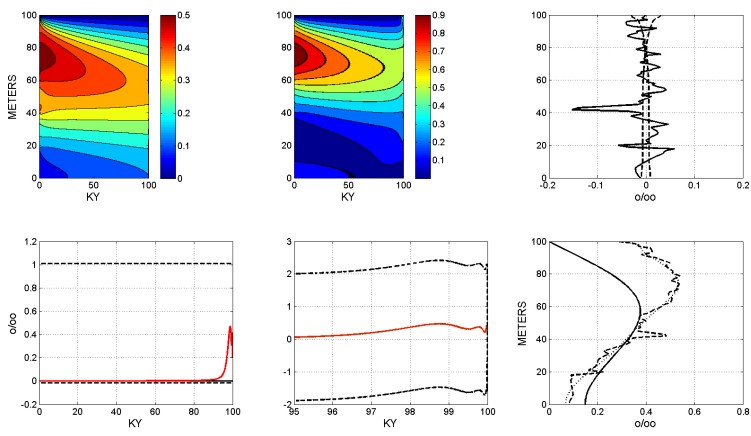

**Figure 15.** Results for Core 981 in the northeast Atlantic Ocean using a flat prior and quasi-periodic boundary conditions for the situation in which the solution is forced to produce the maximum at depth by uncertainty variance weighting.

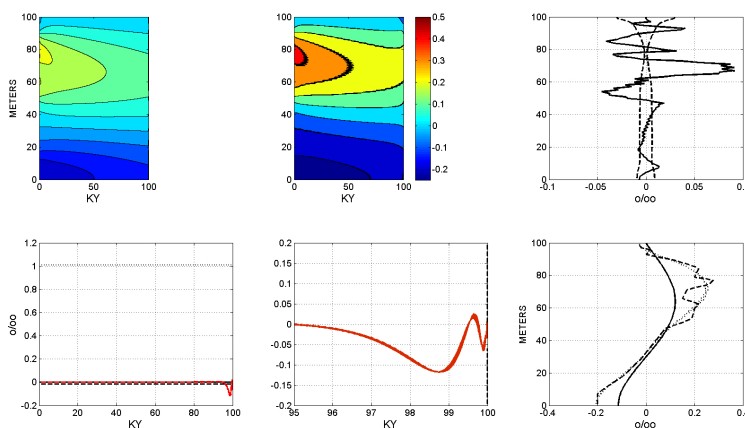

**Figure 16.** Results for Core 1123, east of New Zealand, for a flat prior, quasi-periodic initial conditions, and forcing to the $\delta^{18}O_w$ peak at depth.



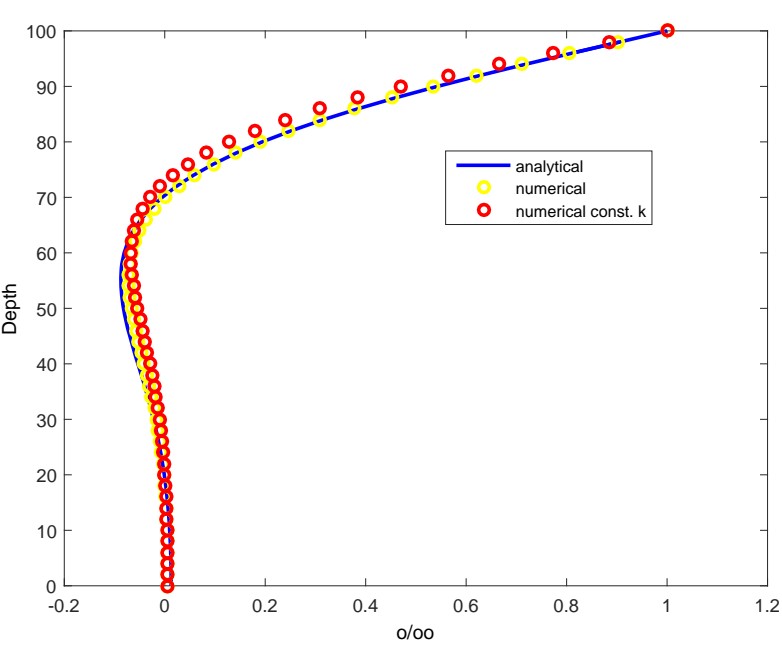

**Figure 17.** Comparison, for a 100 m deep core, of the numerical solution with $k_0 = 10^{-9}$ m$^2$/s, $k_1 = 10^{-11}$ m/s and a 20,000 y period, with the analytical solution (A2). The analytical solution with constant $k = 10^{-9}$ m$^2$/s is also shown.