# Peer review of "Last Glacial Maximum and Deglacial Abyssal Seawater Oxygen Isotopic Ratios"

_Climate of the Past, 2015_

## Referee Comment (RC1) · O. Marchal (Referee) · 9 Feb 2016

This manuscript (ms) presents an analysis of pore water $\delta^{18}$O data from four deep-sea sediment cores (Feni Frift, Bermuda Rise, SW Indian Ridge, and Chatham Rise). Sequential methods of estimation and control theory (a Kalman filter and a related smoother) are used in order to infer, from a quantitative combination of the pore water data with a model describing $\delta^{18}$O transport in pore fluid, the temporal changes in bottom water $\delta^{18}$O overlying the cores over the past 100 kyrs. It is concluded that the presence of bottom waters with very high $\delta^{18}$O during the LGM is possible but not required by the data and the model.

This is a significant contribution. Pore water $\delta^{18}$O and chlorinity data have traditionally been interpreted to mean that ocean bottom waters during the LGM were both very saline (i.e., exceeding salinities expected from estimated volume of land ice) and very cold (i.e., close to freezing point, a second inference that also involves $\delta^{18}$O measurements on benthic foraminifera) (e.g., Adkins et al. 2002). The present analysis shows that more data and a better understanding of tracer transport in pore fluids are needed in order to solidify these traditional inferences of glacial oceanography. It is a welcome addition to the study of Miller et al. (2015), who used a different data analysis method that may be considered by some as being more opaque. It echoes results from this previous study as well as from a recent analysis of chlorinity data by the same author (Wunsch 2015). I recommend publication of the present ms, provided that comments (1-4) below can be addressed or are at least discussed in a revised version. More specific points are listed at the end of this report.

MAJOR COMMENTS

1) As in previous work on this topic, the present analysis requires an assumption about the initial profile of $\delta^{18}$O (here, at t = 100 kyr). In some of the calculations reported in the ms, the initial profile is taken as the measured profile. Whereas this approach appears sensible, it does seem to violate one of the assumptions of the Kalman filter, i.e., the assumption that the errors in the state (here the core $\delta^{18}$O profile) extrapolated from the model and the errors in the data are uncorrelated at every time (e.g., Bryson and Ho 1975, Applied Optimal Control, Taylor & Francis, 1975; transition from eq. (12.2.10) to eq. (12.2.11) on p. 350 of that textbook). Specifically, if the states extrapolated from the model are calculated from initial conditions that are constrained from terminal data, as done here, then the errors in the extrapolated state for the terminal time and the errors in the terminal data are expected to show some correlation. The author is well aware of the assumption of independence between extrapolated state errors and data errors in the Kalman filter (see, e.g., Wunsch 2006; p. 196). Unless my interpretation of the filter's assumptions is incorrect, I would suggest that the apparent violation of this assumption in the present analysis be addressed or at least discussed in the ms.

2) The present study assumes, again as in previous work, that the $\delta^{18}$O flux at the core bottom vanishes. The author is quite upfront with this assumption (p. 3, last paragraph). However, as also acknowledged in the ms (p. 3, bottom), the data do not provide evidence for a vanishing vertical gradient of $\delta^{18}$O and hence of a vanishing vertical $\delta^{18}$O flux at depth in the cores (fig. 3). As stated in the ms, the problem could be reformulated to determine the $\delta^{18}$O fluxes at the core bottom, instead of the core top $\delta^{18}$O values, from the pore water $\delta^{18}$O data (p. 3, last paragraph). I think that the present study would be even more interesting if it also investigates this other problem, i.e., whether

the pore water $\delta^{18}O$ data could be explained by changes in $\delta^{18}O$ flux (or $\delta^{18}O$ value) at the core bottom rather than by changes in $\delta^{18}O$ value at the core top. In fact, that this could be the case is unclear to this reviewer, since the downward effective velocity induced by the postulated decrease of vertical diffusivity with core depth would compete with vertical diffusion in transmitting upward, along the core, information at the core bottom. It would be useful to test whether this intuition is quantitatively grounded on the 100-kyr time scale in a future version of the manuscript.

3) As mentioned in the above comment, the present analysis assumes that the vertical diffusivity of $\delta^{18}O$ (call it kappa) decreases linearly with depth along the cores. A vertical gradient in kappa induces a vertical effective velocity (p. 2), in this case a movement of $\delta^{18}O$ down-core. This movement tends to propagate downwards the information provided by boundary conditions at the core top. As a consequence, it should exert some influence on the controllability of the system and on the results of this study, although the ms suggests that a uniform kappa would make little difference (p. 4, top). I think that a future version of the ms should clarify the basis for the assumption of a decrease of kappa with depth along the cores. For example, is the assumption based on data of sediment porosity and (or) tortuosity? The paper of Wunsch (2015) does report the measured vertical profiles of porosity for the sediment cores, but whether these measurements truly require kappa to decrease with depth is unclear since kappa also depends on other sediment properties such as tortuosity.

4) Most of the calculations reported in the ms seem to assume that the data error variance (**R**) is "about 10 times larger than the value in Adkins and Schrag (2001)" (p. 9). Could this assumption be justified? If data error variance is poorly understood, I would recommend that calculations with different **R** (i.e., with different data errors for all terminal data, not only for data near the measured $\delta^{18}O$ maximum) be conducted in order to further test the robustness of the results.

SPECIFIC POINTS

Abstract and everywhere in the manuscript: replace "salinity/chlorinity" with "salinity (chlorinity)".

Line 4: "… by them."

p1, line 12: "… that the deep ocean …"

p1, line 16: "Recently, Miller (2014) and Miller et al. (2015) have …"

p1, line 20: "… Adkins and Schrag (2003), Miller (2014), and Miller et al. …"

p2, line 4: "… and the model …"

p2, eq. (1b): There should be a minus (not plus) sign in front of the last term on the left-hand side.

p3, line 1: I think that symbols for chemical elements (here oxygen) are generally not italicized.

p3, line 9: "… is an estimate of their uncertainty".

p3, 1$^{st}$ full paragraph: the last sentence may need to be rephrased (a verb seems to be lacking).

p3, line 24: "… based upon measurements in corals …" (sea level curves are not based on $\delta^{18}O$ values in corals).

p4, line 4: "… to the results (see the Appendix)."

p4, line 19: "… matrices **A**, **B**, …" (drop comma after "matrices").

p5, line 16: "Lagrange multipliers, or adjoint, methods and the Rauch…"

p5, line 24: I think that the observability matrix for the present system would be more conventionally defined as the partitioned matrix (e.g., Gelb et al. 1974):

$$O = [0|0| \dots |(A^T)^{t_f-1}I_{2M}],$$

which has a similar form as the controllability matrix (5).

p6, line 9: "… 100,000 yr …"

p6, line 24: "…$\delta^{18}O_w$ distribution …" (drop comma).

p7, "… likely connected to the extreme volatility of dynamical properties in the equatorial Pacific Ocean and is not further discussed here": this could be elaborated or dropped.

p7, line 17, and everywhere in the manuscript: replace "… and/or …" with "… and (or) …".

p8, line 18: please define the matrices $P_0$, **Q**, and **I**.

p8, line 20: "… k decreases linearly with core depth from … at z = h to … at z = 0 m."

p9, line 3: "… conditions, figure 8 shows …"

p9, line 11: "… and the terminal data uncertainty was strongly reduced in the vicinity …"

p9, lines 22-23: "… (about 10 times .. in Adkins and Schrag [2001] and now meant … error), $P_0$ = …".

p9, last sentence: is it meant "It would appear that this record is not consistent with the prior d18O profile at 100 kyr within its stipulated error bars"? Some clarification would be useful.

p10, section 3.3: Please also explore cases with a sea-level prior and zero initial conditions, and briefly describe the results in the ms.

p10, section 3.4: Please also explore the case with zero initial conditions and briefly describe the results in the ms.

p10, line 23: "… rough summary would include: …"

p10, line 4: "(1) Physical transport of $\delta^{18}O$ is one-dimensional (vertical)". Although this seems to be common in the literature, I would suggest not to use the division sign in non-mathematical expressions, such as "Physics/chemistry", "diffusivity/porosity", "Advection/diffusion", etc.

p11, line 23 (in Appendix): "… from the bottom of the core at z = 0 m to the top of the core at z = h". In the development following (A1), I would suggest first to introduce the change in variable ($\zeta =

…"$) and then to set $c = \hat{c}(\zeta) …$". The ordinary differential equation between (A1) and (A2) should have $\zeta$ as the sole independent variable.

p14, line 16 (Reference list): have "Olver …" starting on a separate line.

Fig. 1 could be enlarged.

Caption of Table 2: "… $\boldmath{x}_t$ …"

Figs. 7-16 could all be enlarged.

Caption of fig. 7: "… with k linearly increasing from … at the core bottom (z = 0 m) to … at the core top (z = h)." In panel (c), I interpret the solid line as the difference between the filter estimate of d18O and the measured $\delta^{18}O$, but I am not sure. Panel (e) could be zoomed in, perhaps on the last kyr, to better see the changing control and its estimated error near t = 0 kyr.

Caption of fig. 10: Please define **R** in the main text.

Fig. 17: the legend indicates that the solution with constant kappa is a numerical one, whereas the caption indicates that it is an analytical one. Please clarify. The initial conditions and the time for which the solutions are displayed could be specified.

Olivier Marchal

---

## Referee Comment (RC2) · Anonymous Referee #2 · 29 Mar 2016

**1  Summary**

In this manuscript, the author extends his salinity-only analysis (Wunsch, 2015) to the pore-water measurements of the oxygen isotope ratio $\delta^{18}O_w$. Using "standard control theory", he arrives at the same conclusion as Müller et al. (2015) – who use a Markov Chain-Monte Carlo (MCMC) approach – and as in his previous work: A very cold, highly saline abyssal ocean during the Last Glacial Maximum (LGM) is possible, but not required by the existing data. This conclusion is important as it gives lesser weight to a constraint that has been imposed on many attempts to reconstruct the deep-ocean circulation during the LGM.

**2 Major comments**

I recommend to (1) add some more detail to the description of the method such that – although short – it can stand it by itself and (2) rephrase or shorten paragraphs that sound (pardon me) a bit like a text book (see below).

P. 5: What is the relevance of the subsections "Identification" and "State Estimation and Control" for the current manuscript? They sound a bit "text book-like". With respect to the identification problem, which assumption is made in the end and what is the concrete solution that is proposed?

P. 6: Is the "fuller discussion of contollability and observability" essential, or is the present discussion sufficient? This is another "text book-like" statement that may confuse the reader.

Are really all Figures 7 to 16 needed, or could one select one or two prominent examples?

**3 Minor comments**

Throughout.the manuscript, the same symbol $\delta^{18}O_w$ should indicate the oxygen isotope ratio of pore water (or seawater). Furthermore, a true percent sign should be used.

P. 3, lines 11-13: [...] and represent the values that any estimate of the core values through time, $c(z,t), 0 < t < t_f$, must converge to within error bars [...]

P. 3, line 18: should refer to Table 2 (p. 16)

P. 4, line 7: introduce abbreviation "RTS" here

P. 16: The caption of Table 2 needs to be checked for the use of LaTeX and punctuation.

In the present version of the manuscript, the figures are generally too small and consequently the text is barely readable, especially the annotation of the axes.

Figure 1 (also in Figures 7 to 16): The horizontal axis should be clearly marked as "Time/(ka BP)" or "Time [ka BP]", and it should start at -100 ka BP (kilo-years before present) to make it easier to the reader to recognize the last glacial cycle and the LGM.

Figure 11: It should probably read $\delta^{18}O_w$ instead of $\delta^{18}O_{sw}$.

Figure 13: not clear which figure one should compare to (Figure 7?)

Figure 14: It should probably read $\delta^{18}O_w$ instead of $\delta^{13}C_w$.

―――――――――――――――――

---

## Author Comment (AC1) · 11 Apr 2016

My thanks to Olivier Marchal for again making very helpful comments. In response, starting with the "Major Comments":

1) As in previous work on this topic, the present analysis requires an assumption about the initial

profile of $\delta^{18}O$ (here, at t = 100 kyr). In some of the calculations reported in the ms, the initial profile

is taken as the measured profile. Whereas this approach appears sensible, it does seem to violate

one of the assumptions of the Kalman filter, i.e., the assumption that the errors in the state (here

the core $\delta^{18}$O profile) extrapolated from the model and the errors in the data are uncorrelated at

every time (e.g., Bryson and Ho 1975, Applied Optimal Control, Taylor & Francis, 1975; transition

from eq. (12.2.10) to eq. (12.2.11) on p. 350 of that textbook). Specifically, if the states extrapolated

from the model are calculated from initial conditions that are constrained from terminal data, as

done here, then the errors in the extrapolated state for the terminal time and the errors in the

terminal data are expected to show some correlation. The author is well aware of the assumption of

independence between extrapolated state errors and data errors in the Kalman filter (see, e.g.,

Wunsch 2006; p. 196). Unless my interpretation of the filter's assumptions is incorrect, I would

suggest that the apparent violation of this assumption in the present analysis be addressed or at

least discussed in the ms.

*The point is correct, that a priori errors correlated in time are not accounted for properly in the basic sequential estimation method. Here, however, one must distinguish between the use of identical data for the initial and final states, and a very different*

*assumption that the corresponding errors are in any way related. I've assumed that the ways in which the initial state would differ from the correct one have an entirely different error structure from the deviation at the terminal state. So, by way of example, terminal errors could be dominantly analytical ones, while deviations at $t = -100$ ky would be dominated by a whole host of processes for which the analytical error at $t = 0$ would likely be completely negligible. (I've added a clarifying sentence.)*

2) The present study assumes, again as in previous work, that the $\delta^{18}$O flux at the core bottom

vanishes. The author is quite upfront with this assumption (p. 3, last paragraph). However, as also

acknowledged in the ms (p. 3, bottom), the data do not provide evidence for a vanishing vertical

gradient of $\delta^{18}$O and hence of a vanishing vertical $\delta^{18}$O flux at depth in the cores (fig. 3). As stated in

the ms, the problem could be reformulated to determine the $\delta^{18}$O fluxes at the core bottom, instead

of the core top $\delta^{18}$O values, from the pore water $\delta^{18}$O data (p. 3, last paragraph). I think that the

present study would be even more interesting if it also investigates this other problem, i.e., whether

the pore water $\delta^{18}$O data could be explained by changes in $\delta^{18}$O flux (or $\delta^{18}$O value) at the core

bottom rather than by changes in $\delta^{18}$O value at the core top. In fact, that this could be the case is

unclear to this reviewer, since the downward effective velocity induced by the postu-
lated decrease

of vertical diffusivity with core depth would compete with vertical diffusion in transmit-
ting upward,

along the core, information at the core bottom. It would be useful to test whether this
intuition is

quantitatively grounded on the 100-kyr time scale in a future version of the manuscript.

*The bottom boundary condition is troublesome. But relaxing it to permit finite vertical
diffusion from below would not add much to what we already know: the result will de-
pend directly upon the assumptions concerning the magnitude and sign of $w$, and the
magnitude of $k$ as well as guesses at the statistics, at least, of the temporal variations
there. I hope that someone will pursue this (I might), but the message of the present
paper already suggests so much freedom in guessing the correct physical situation
that I am loathe to explore yet another one.*

3) As mentioned in the above comment, the present analysis assumes that the vertical
diffusivity of

$\delta^{18}$O (call it kappa) decreases linearly with depth along the cores. A vertical gradient
in kappa induces

a vertical effective velocity (p. 2), in this case a movement of $\delta^{18}$O down-core. This
movement tends

to propagate downwards the information provided by boundary conditions at the core
top. As a

consequence, it should exert some influence on the controllability of the system and
on the results

of this study, although the ms suggests that a uniform kappa would make little difference (p. 4, top).

I think that a future version of the ms should clarify the basis for the assumption of a decrease of

kappa with depth along the cores. For example, is the assumption based on data of sediment

porosity and (or) tortuosity? The paper of Wunsch (2015) does report the measured vertical profiles

of porosity for the sediment cores, but whether these measurements truly require kappa to

decrease with depth is unclear since kappa also depends on other sediment properties such as

tortuosity.

*Numerical experiments, not shown, demonstrate that the "induced" vertical velocity only quantitatively modifies the results for these values of $k$. A full discussion of the physics governing advection/diffusion in a core, including such zero-order issues as the utility of the one-dimensional assumption, would be a major undertaking for someone more fully competent in flows in porous media at high pressures. I also added some words about the assumptions concerning the sediment-water interface physics.*

4) Most of the calculations reported in the ms seem to assume that the data error variance (R) is

"about 10 times larger than the value in Adkins and Schrag (2001)" (p. 9). Could this assumption be

justified? If data error variance is poorly understood, I would recommend that calculations with

different R (i.e., with different data errors for all terminal data, not only for data near the measured

$\delta^{18}$O maximum) be conducted in order to further test the robustness of the results.

*Such experiments have been done, but don't really change anything. I've added some sentences about sensitivity to terminal data errors—where the major issue, not resolved, is whether the observed structures are signals or noise.*

SPECIFIC POINTS

Abstract and everywhere in the manuscript: replace "salinity/chlorinity" with "salinity (chlorinity)". *Ok*

Line 4: "... by them."*Ok*

p1, line 12: "... that the deep ocean ..." *Ok*

p1, line 16: "Recently, Miller (2014) and Miller et al. (2015) have ..." left.

p1, line 20: "... Adkins and Schrag (2003), Miller (2014), and Miller et al. ..."*Ok*

p2, line 4: "... and the model ..." *Ok*

p2, eq. (1b): There should be a minus (not plus) sign in front of the last term on the left-hand side. yes

p3, line 1: I think that symbols for chemical elements (here oxygen) are generally not italicized. *Ok*

p3, line 9: "... is an estimate of their uncertainty". *Ok*

p3, 1st full paragraph: the last sentence may need to be rephrased (a verb seems to be lacking). ?*I think ok?*

p3, line 24: ". . . based upon measurements in corals . . ." (sea level curves are not based on f06418O values

in corals). *ok*

p4, line 4: ". . . to the results (see the Appendix)." *Ok*

p4, line 19: ". . . matrices A, B, . . ." (drop comma after "matrices"). *Ok*

p5, line 16: "Lagrange multipliers, or adjoint, methods and the Rauch. . ." *Ok*

p5, line 24: I think that the observability matrix for the present system would be more

conventionally defined as the partitioned matrix (e.g., Gelb et al. 1974):

= [0|0| . . . ( ) ],

which has a similar form as the controllability matrix (5). *True, but I've written a special case, now noted.*

p6, line 9: ". . . 100,000 yr . . ."*Ok*

p6, line 24: ". . . f06418Ow distribution . . ." (drop comma). *Ok*

p7, ". . . likely connected to the extreme volatility of dynamical properties in the equatorial Pacific

Ocean and is not further discussed here": this could be elaborated or dropped. *Disagree. MS. explains why not further discussed.*

p7, line 17, and everywhere in the manuscript: replace ". . . and/or . . ." with ". . . and (or) . . .". *A journal style choice. I will wait and see.*

p8, line 18: please define the matrices P0, Q, and I. *Dropped.*

p8, line 20: ". . . k decreases linearly with core depth from . . . at z = h to . . . at z = 0 m." *Ok*

p9, line 3: "... conditions, figure 8 shows ..." *Ok*

p9, line 11: "... and the terminal data uncertainty was strongly reduced in the vicinity ..." *Ok*

p9, lines 22-23: "... (about 10 times .. in Adkins and Schrag [2001] and now meant ... error), P0 = ...". *Dropped*

p9, last sentence: is it meant "It would appear that this record is not consistent with the prior d18O *Dropped*

profile at 100 kyr within its stipulated error bars"? Some clarification would be useful.

p10, section 3.3: Please also explore cases with a sea-level prior and zero initial conditions, and

briefly describe the results in the ms. *Too many cases already!*

p10, section 3.4: Please also explore the case with zero initial conditions and briefly describe the

results in the ms. *Same as above*

p10, line 23: "... rough summary would include: ..."*Ok*

p10, line 4: "(1) Physical transport of f06418O is one-dimensional (vertical)". Although this seems to be

common in the literature, I would suggest not to use the division sign in non-mathematical

expressions, such as "Physics/chemistry", "diffusivity/porosity", "Advection/diffusion", etc. *Again a journal style decision. I don't think much danger of confusion.*

p11, line 23 (in Appendix): "... from the bottom of the core at z = 0 m to the top of the core at z = h". *Ok.*

In the development following (A1), I would suggest first to introduce the change in variable ($\zeta =

...$") and then to set $c = \hat{c}(\zeta) ...$". The ordinary differential equation between (A1) and (A2)

should have $\zeta$ as the sole independent variable. *yes.*

p14, line 16 (Reference list): have "Olver ..." starting on a separate line. *Ok*

Fig. 1 could be enlarged. *Ok*

Caption of Table 2: "... $\boldmath{x}_t$ ..." *ok*

Figs. 7-16 could all be enlarged. *Yes*

Caption of fig. 7: "... with k linearly increasing from ... at the core bottom (z = 0 m) to ... at the core

top (z = h)." In panel (c), I interpret the solid line as the difference between the filter estimate of

d18O and the measured f06418O, but I am not sure. *Fixed*

Panel (e) could be zoomed in, perhaps on the last

kyr, to better see the changing control and its estimated error near t = 0 kyr. ??

Caption of fig. 10: Please define R in the main text. *Ok*

Fig. 17: the legend indicates that the solution with constant kappa is a numerical one, whereas the

caption indicates that it is an analytical one. Please clarify. The initial conditions and the time for

which the solutions are displayed could be specified. *Ok. Fixed*

Olivier Marchal

---

## Author Comment (AC2) · 11 Apr 2016

In this manuscript, the author extends his salinity-only analysis (Wunsch, 2015) to the pore-water measurements of the oxygen isotope ratio 18Ow. Using "standard control theory", he arrives at the same conclusion as Müller et al. (2015) – who use a Markov Chain-Monte Carlo (MCMC) approach – and as in his previous work: A very cold, highly saline abyssal ocean during the Last Glacial Maximum (LGM) is possible, but not required by the existing data. This conclusion is important as it gives lesser weight to a constraint that has been imposed on many attempts to reconstruct the deep-ocean circulation during the LGM.

2 Major comments

I recommend to (1) add some more detail to the description of the method such that
– although short – it can stand it by itself and (2) rephrase or shorten paragraphs that
sound (pardon me) a bit like a text book (see below).

P. 5: What is the relevance of the subsections "Identification" and "State Estimation and
Control" for the current manuscript? They sound a bit "text book-like". With respect
to the identification problem, which assumption is made in the end and what is the
concrete solution that is proposed?

P. 6: Is the "fuller discussion of contollability and observability" essential, or is the
present discussion sufficient? This is another "text book-like" statement that may confuse
the reader.

*The puzzle facing any writer of a paper such as this one is to figure out who the audience is likely to be—and whether the mathematical bits will put off the geochemists/geologists, and (or) vice-versa? One runs the risk of falling between two stools. I've now tried to explain in the text e.g., why "Identification" and "State Estimation and Control" are mentioned—they are general pieces of machinery that can be used to evaluate a system in advance of the data, among other possibilities.*

Are really all Figures 7 to 16 needed, or could one select one or two prominent examples. *The other reviewer called for more examples! Given the very large number of possibilities, I've decided to drop one of the figures, and condense some of the remaining discussion.*

[Figure]

3 Minor comments

Throughout.the manuscript, the same symbol 18Ow should indicate the oxygen isotope
*Ok*

ratio of pore water (or seawater). Furthermore, a true percent sign should be used.
*Will leave to publisher.*

P. 3, lines 11-13: [...] and represent the values that any estimate of the core values

through time, c(z; t); $0 < t < $ tf , must converge to within error bars [: : :]*Ok*

P. 3, line 18: should refer to Table 2 (p. 16) *Ok*

P. 4, line 7: introduce abbreviation "RTS" here *Ok*

P. 16: The caption of Table 2 needs to be checked for the use of LaTeX and punctuation.
*Yes*

In the present version of the manuscript, the figures are generally too small and con-
sequently

the text is barely readable, especially the annotation of the axes. *Fixed subject to
display size.*

Figure 1 (also in Figures 7 to 16): The horizontal axis should be clearly marked as

"Time/(ka BP)" or "Time [ka BP]", and it should start at -100 ka BP (kilo-years before

present) to make it easier to the reader to recognize the last glacial cycle and the LGM.
*Done*

Figure 11: It should probably read 18Ow instead of 18Osw. *Ok*

Figure 13: not clear which figure one should compare to (Figure 7?) *Ok*

Figure 14: It should probably read 18Ow instead of 13Cw. *Ok*

[Figure]

---

## Author Response (AR1)

Response to Editor's Final Remarks:

The request was for saying something about the actual accuracies of $\delta^{18}O_c$ in the context of the possibility of below-freezing temperatures at the LGM. I have added three sentences P. 2, line 19, being more explicit about the problem. But I've re-emphasized that the goal here is to ask whether the pore water values alone require the high values, rather than being an inference about the carbonate values themselves. The issue is briefly recapitulated in the final discussion paragraph.

I hope all is now acceptable.

CW